# Effects of Problem-Based Learning Strategies on Undergraduate Nursing Students’ Self-Evaluation of Their Core Competencies: A Longitudinal Cohort Study

**DOI:** 10.3390/ijerph192315825

**Published:** 2022-11-28

**Authors:** Yen-Chiao Angel Lu, Shu-Hsin Lee, Ming-Yi Hsu, Fen-Fen Shih, Wen-Jiuan Yen, Cheng-Yi Huang, Pei-Ching Li, Ching-Yen Hung, Hsiao-Ling Chuang, Ching-Pyng Kuo

**Affiliations:** 1Department of Nursing, Chung Shan Medical University, Taichung 40201, Taiwan; 2Department of Nursing, Chung Shan Medical University Hospital, Taichung 40201, Taiwan

**Keywords:** problem-based learning, PBL, self-evaluation, core competency, nursing student

## Abstract

To respond to patients’ increasing demands and strengthen nursing professionals’ capabilities, nursing students are expected to develop problem-solving skills before they enter the workforce. Problem-based learning (PBL) is expected to provide effective simulation scenarios and realistic clinical conditions to help students achieve those learning goals. This article aims to explore the effects of PBL strategies on nursing students’ self-evaluation of core competencies. This longitudinal cohort survey study evaluated 322 nursing students attending Chung Shan Medical University, Taiwan, in 2013 and 2014, where PBL teaching strategies are used in all four undergraduate years from freshman to senior. Based on their undergraduate academic levels, students were categorized into three groups- one-year PBL exposure, two-year PBL exposure, and three-year exposure. A core competency questionnaire was administered twice to ask participants to self-assess five professional competencies: learning attitude, problem identification, information analysis, execution, and life-long learning. The results showed that students with the longest exposure to PBL (Group 3) had higher self-evaluated scores for all core competencies than the other groups, except for the execution competency. The mean total competency score increased by 0.12 points between the pre-and-test. In addition, the mean score increased significantly more in Group 3 than in Groups 1 and 2. These trends were consistent for the information analysis, execution, and life-long learning competencies. In conclusion, the changes in the self-evaluated scores between groups indicate PBL strategies effectively improve nursing students’ core competencies. The longest exposure group reported higher self-evaluated core competency scores than the other groups, especially for the information analysis, execution, and life-long learning competencies.

## 1. Introduction

In recent years, improvements in teaching and learning strategies have led to continual expansion of the teaching methods available to educators [1,2]. Additionally, nursing curricula should be designed to prepare nursing students for their future careers. Therefore, nursing educators have emphasized a shift from traditional teaching methods to learner-centered teaching strategies to improve the effectiveness of learning [2,3]. To respond to the increasing demands of patients and strengthen nursing professionals’ capabilities, nurses are expected to develop problem-solving skills [4,5]. However, gaps still exist between the curricula of current nursing education programs and the requirements of clinical practice [6,7]. Moreover, due to the increasing complexity and demands of healthcare, nurses are expected to develop critical thinking skills to assess complex clinical problems [8,9] and provide holistic care [10]. Thus, it is necessary to adopt new methodologies for nursing education to respond to the rapidly changing medical context and promote key competencies in the areas of problem-solving, critical thinking, and creativity among nurses.

During the process of integrating nursing students into clinical situations, case problems should be based on realistic clinical conditions [11] and challenge nursing faculty. Therefore, the incorporation of PBL learning provides a strategy to flip the classroom [12] and can provide effective simulation scenarios and parallel learning experiences.

The problem-based learning (PBL) strategy was developed by the Faculty of Health Sciences of McMaster University in the late 1960s [13]. The PBL strategy has since been adopted in numerous medical [14,15] and nursing schools [16,17], including in Taiwan [18,19,20]. PBL is a student-centered approach that aims to facilitate autonomous and independent learning and seeks to enable students in higher education to apply knowledge, acquire skills, and achieve diverse learning goals [21]. Compared with nursing students who study traditional education courses, this approach develops self-directed learners, who build capability in terms of critical thinking, leadership, and teamwork [21]. Therefore, the role of the educator in PBL is entirely considered as a facilitator and to allow learners to seek a variety of potential solutions [22]. PBL strategies have been designed to promote learner interaction, problem-solving, and learning through teamwork [23]. Educators develop various learning activities based on actual situations and case problems to encourage autonomous learning [23,24]. Research has shown that PBL teaching strategies can effectively improve students’ scores for curiosity, systematic, analytical, and critical thinking [25], as well as their leadership, teamwork [21], and problem-solving abilities [26].

Despite PBL being widely used in professional healthcare courses, few studies have assessed the effectiveness of PBL using valid instruments or employed longitudinal follow-up to measure the strength of the effects. Furthermore, most related studies were qualitative [27,28], only measured the performance of some abilities [3,5,29], or focused on evaluations of student achievements by faculty [30]. Therefore, this study aimed to evaluate the influence of PBL strategies on students’ self-evaluation of their core competencies over two years among students studying at different grades of the same nursing school in Taiwan.

## 2. Methods

### 2.1. Research Design and Subjects

This study adopted a longitudinal survey design to investigate the differences in nursing students’ self-evaluations of their core competencies over two years between grades.

At Chung Shan Medical University, PBL teaching strategies are used in the course design of various nursing modules (from freshman to senior). In September 2013, we recruited freshman to junior class nursing students attending Chung Shan Medical University for this study. In September 2014, these students were advanced to sophomore, junior and senior standing. Overall, a total of 322 nursing students completed the same questionnaire in both 2013 and 2014.

We grouped the students by their undergraduate academic level to study the effect of exposure to the PBL strategy: Group 1 (*n* = 106, 2013 as Freshman students, 2014 as Sophomore students) had PBL exposure for 1 year; Group 2 had PBL exposure of 2 years (*n* = 111, 2013 as Sophomore students, 2014 as Junior students); and Group 3 had PBL exposure of 3 years (*n* = 105, 2013 as Junior students, 2014 as Senior students).

### 2.2. Measures

Data on the students’ characteristics were collected, including age, gender, and grade level.

A core competencies questionnaire was designed based on a literature review and nursing education principles as a framework. Content validity was performed, and five dimensions were identified: learning attitude, problem identification, information analysis, execution, and life-long learning. The overall content validity index of the instrument was high (overall S-CVI = 0.93) The final instrument includes sixteen items out of twenty-four items originally developed. Individual items related to the five dimensions were provided in Table 1.

“Learning attitude” was defined as students’ tendency to respond a certain way towards PBL learning. Four items were included in the questionnaire to measure students’ learning attitudes. “Problem identification” was students’ ability to identify the key problem to be solved or addressed. Two items were used to assess subjects’ problem-identification skills. “Information analysis” aimed to check students’ ability to inspect and comprehend information collected. Three items were included to evaluate students’ information analysis competency. “Execution” focused on students’ abilities to perform a certain task or complete study goals. Three items were employed to assess students’ execution competency. “Life-long learning” was described as students’ self-initiated and ongoing learning intentions or behaviors. It was measured by 4-item questions.

A five-point Likert scale was adopted for data collection and the responses were rated as follows: 1 = Strongly disagree, 2 = Disagree, 3 = Neutral (neither agree nor disagree), 4 = Agree, and 5 = Strongly agree. A higher score indicates a positive self-evaluation of core competencies.

Students (from freshman to junior classes) were recruited to participate in the survey in September 2013. Those students were tracked to complete the survey in September 2014. Overall, a total of 322 nursing students completed the same questionnaire in both 2013 and 2014.

Each student was asked to self-evaluate their core competencies using the same questionnaire in 2013 and 2014. The Cronbach’s alpha values of internal consistency were 0.944 for the overall scale and 0.772 to 0.866 for each of the five dimensions (learning attitude, 0.772; problem identification, 0.844; information analysis, 0.821, execution, 0.777; life-long learning, 0.866). Therefore, the present study had acceptable internal consistency.

### 2.3. Data Collection and Analysis

We collected data twice during the research period, as all participants completed the same core competencies questionnaire in both 2013 and 2014. The Institutional Review Board approved this study at the Chung Shan Medical University Hospital (No: CS2-21113), and data were only collected after informed consent had been obtained. All participants signed an informed consent stating that they had the right to withdraw from the study at any time. The data collected were only used for research purposes.

Descriptive statistics including frequencies, percentages, means, and standard deviations were used to assess the distributions of the students’ characteristics and core competencies. Chi-square tests and paired *t*-tests were used to analyze the differences between groups. Radar charts were plotted to compare the changes in the student’s self-evaluations of their core competencies between groups. To adjust for the interaction between exposure and group, we used generalized estimating equations and statistical methods to predict the changes in the core competencies of each group. The significance level was set at *p* < 0.05 and all tests were two-tailed. The SPSS for Windows version 20.0 (SPSS Inc., Chicago, IL, USA) was used for data analysis.

## 3. Results

Descriptive statistics such as Chi-square tests, paired-sample *t*-tests, and generalized estimating equations (GEE) were used to analyze data.

### 3.1. Characteristics of the Nursing Students

A total of 322 nursing students (Group 1 = 106, Group 2 = 111, and Group 3 = 105 students) completed the core competency questionnaire twice, in 2013 and 2014. Though most students were female (81.4%), Chi-square tests confirmed that there was no significant difference (*p* = 0.379) in the sex distribution between groups, showing that the distribution of the characteristics was homogenous between groups (Table 2).

### 3.2. Comparison of the Differences in Students’ Self-Evaluations of Core Competencies between Groups

Pre-and post-tests revealed significant changes (from 2013 to 2014) in the total self-evaluated competency scores of the students in Group 3 (*p* < 0.000). Similarly, significant differences were observed for the competencies learning attitude (*p* = 0.007), problem identification (*p* = 0.0181), information analysis (*p* = 0.007), and life-long learning (*p* < 0.000) in Group 3. However, the score for the execution competency (*p* = 0.086) did not significantly change in Group 3. In Group 2, pre-and post-tests showed the students’ self-evaluation scores for all competencies did not significantly change. In Group 1, students’ self-evaluation scores were only significantly different for the execution (*p* = 0.021) and life-long learning competencies (*p* = 0.037) in pre-and post-tests (Table 3).

Furthermore, we used radar charts to compare the changes in the students’ self-evaluations of their core competencies between groups. The students in group 3 had higher self-evaluation scores for the core competencies than the students in the other groups, except for the execution competency. In Group 1, the most significant change was observed in the competency of execution. The changes in student scores were lowest in Group 2; only the problem identification competency score was significantly higher in Group 2 compared with Group 1 (see Figure 1).

### 3.3. Predicted Changes in the Students’Sself-Evaluations of Their Core Competencies

The study questionnaire was administered twice to the same students, in 2013 and 2014. Thus, a GEE model was used to analyze the trends in the changes in the self-evaluated core competency scores between the three groups of nursing students over time. The mean changes in the total core competency scores between groups remained significantly different after controlling for the interaction effect between exposure (i.e., years of exposure to PBL) and group.

We found that the mean self-evaluated total competency score increased by 0.12 points at the post-test compared with the pre-test (*p* = 0.038). Moreover, the mean total score (five competencies) for Group 3 significantly increased by 0.286 points compared with Group 1 (*p* < 0.000), and the mean score of Group 2 also significantly increased by 0.199 points compared with Group 1 (*p* = 0.006; Table 4).

The students’ mean post-test scores for the competency information analysis, execution, and life-long learning increased by 0.151–0.175 points compared with the pre-test score (*p* = 0.013–0.032). In addition, the mean score of Group 3 for these competencies significantly increased by 0.262–0.352 points compared with Group 1 (*p* < 0.000–0.006), and the mean score of Group 2 significantly increased by 0.183–0.264 points compared with Group 1 (*p* = 0.002–0.034; Table 4).

However, after controlling for the interaction of exposure and group, there were no significant differences between the mean pre-and post-test scores for the competencies of learning attitude and problem identification (*p* = 0.154–0.343). However, the mean learning attitude score of Group 3 significantly increased by 0.223 points compared with Group 1 (*p* = 0.004), and the mean learning attitude score of Group 2 significantly increased by 0.159 points compared with Group 1 (*p* = 0.033). However, the mean problem identification competency score was only significantly different between Group 3 compared with Group 1, with an increase of 0.347 points in Group 3 (*p* < 0.000; Table 4).

## 4. Discussion

The study aimed to explore the effects of PBL strategies on nursing students’ self-evaluation of core competencies. Nursing educators needed to evaluate the effects of PBL teaching strategies on students. These study results would help nursing educators develop curricula and improve teaching strategies. We found that the students with the longest exposure to PBL (Group 3) had more significant improvements in their self-evaluation scores for all competencies than the shorter exposure groups (groups 1 and 2). Thus, this study demonstrates that nursing students’ self-evaluations of their core competencies improved each grade, which indicates PBL strategies effectively help students build core competencies. The PBL model advocates that a paradigm shift is required in nursing education [3]. Yet, despite the number of teaching experiences reported to date with PBL, almost no studies in Taiwan have assessed self-evaluations of the core competencies of students educated using this strategy. Most research has focused on the impact of PBL on individual capabilities, including critical thinking [29,31], problem-solving [26], and metacognitive awareness [31]. Other studies have focused on learner satisfaction [32,33]. This study found that nursing students studying a PBL curriculum gradually improved their total core competencies as their exposure to PBL increased. Specifically, our study shows that students’ execution and life-long learning competencies were improved by the end of the sophomore year. In addition, the students’ self-evaluations of the learning attitude, problem identification, and information analysis competencies were significantly improved by the end of their junior year. Researchers have indicated that the PBL strategy encourages students to discuss and cooperate to solve problems through self-directed study [21,28,34]. Cultivation of these abilities is necessary to accumulate knowledge and, combined with exposure to clinical situations, prepare nursing students to adapt to the reality of the workplace [3,28].

Learning strategies are essential components of a curriculum and could help students to learn more efficiently and effectively [13]. In addition, new learning strategies should emerge spontaneously to address the increasingly complex clinical environment and rapidly changing clinical patient problems [30,35]. Therefore, PBL strategies are one option [10,36]. Many studies have recently discussed the effectiveness of this learning strategy in health discipline educational programs [6]. However, how students self-evaluate their development of core competencies in nursing education was poorly understood [37]. Therefore, more research from the perspective of nursing students’ self-evaluations of their progression in terms of core competencies is necessary to confirm the effectiveness of PBL strategies [37,38]. This study showed that exposure to PBL strategies improved the students’ self-evaluations of their core competencies, and the differences between grades and individual progression between years were significant.

Research has indicated that the strengths of PBL include its ability to promote the integration of knowledge, problem-solving skills, critical thinking skills, group collaboration, and self-autonomous learning [13,30,39]. Our students experienced a series of PBL teaching courses and their self-evaluation scores revealed that PBL increased the students’ perceptions of their core competencies. The improvements in the information analysis, execution, and life-long learning competencies were directly proportional to the grade of the students, and hence the duration of their exposure to PBL. However, the scores for the core competencies of learning attitude and problem identification only increased significantly after the junior year, indicating that the PBL teaching strategy could lead to the accumulation of skills, especially in the learning attitude and problem identification competencies. Thus, the results of this study show that PBL teaching provides a strategy that allows students to become self-directed learners and cooperate with other members of a team. The benefits of the PBL teaching strategy were previously demonstrated by students’ responses in three areas of cognitive, emotional, and social skills [40].

Despite PBL being widely used in professional healthcare courses, few studies have assessed the effectiveness of PBL using specific instruments. Most studies have only assessed outcomes based on qualitative data [27,28], students’ ability to perform a single skill [3,5,41], learner satisfaction [12,42], or evaluations by faculty or the academic performance of students [30,43]. The present study used an instrument to assess the effectiveness of PBL by measuring nursing students’ self-evaluation of their competency in five domains: learning attitude, problem identification, information analysis, execution, and life-long learning. Thus, in contrast to other studies that only assessed a single ability or where only faculty evaluation of students’ performance was used, our study represents a more comprehensive program evaluation method that includes both faculty evaluation and students’ self-assessment.

Overall, this study indicates PBL has cumulative effects on core competencies as nursing students progress through their academic years. Therefore, teaching strategies could be designed according to the grade and maturity of students, as well as the sequence of formation of core competencies. PBL teaching strategies can provide students with experience of actual cases and promote deep self-learning and the development of core competencies. Using this approach, nursing students will be able to adapt to the changing clinical environment of the future and solve diverse health problems.

## 5. Conclusions

This study highlights the ability of PBL strategies to promote the development of core competencies in an undergraduate nursing course. Therefore, PBL appears to have a favorable effect on nursing education. The differences in the self-evaluation scores between groups indicate PBL strategies effectively improve nursing students’ core competencies. The students with the longest exposure to PBL had higher self-evaluation scores than the other groups, especially for competency information analysis, execution, and life-long learning. The difference in progress between groups was most obvious in Group 3, which suggests that a PBL curriculum design has a cumulative effect on students’ development of core competencies.

However, this study only sampled students studying at a single school using the same curriculum and measured the student’s self-evaluations of their core competencies using a single questionnaire. Thus, the results of this study may not be generalizable to other schools or curricula. Consequently, it is necessary to expand this curricular design to other schools and assess a larger sample size to evaluate the effectiveness of PBL teaching strategies and further research is recommended to confirm the reliability and validity of the core competency measurement tool used in this study. A follow-up outcome study will be conducted to examine the long-term effects of PBL on the core competencies of graduates. We believe that extending the time for data collection would enable an analysis of the trajectory of the long-term effects and provide more specific data on the effectiveness of PBL.

## Figures and Tables

**Figure 1 ijerph-19-15825-f001:**
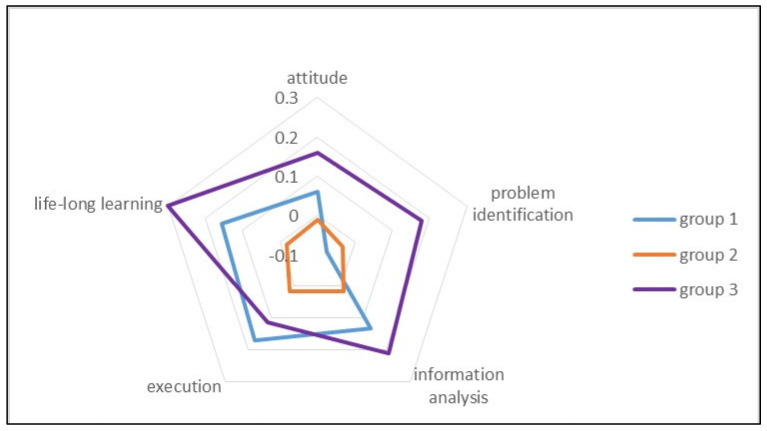
The radar chart comparing the changes in the core competencies.

**Table 1 ijerph-19-15825-t001:** Items analyzed in the questionnaire.

Dimensions	Number of Items	Items Indicators
Learning Attitudes	4	I like solving problems regarding the group project with team members. I enjoy cooperative learning with team members. When I encounter a problem, I learn to think, understand and analyze the problem first. I have learned to tolerate and respect different opinions.
Problem Identification	2	I can identify problems from the case scenarios provided. I am familiar with problem-solving techniques.
Information Analysis	3	It is easy for me to find the appropriate references to meet course objectives. I can evaluate the credibility and reliability of data or information retrieved. I can conclude from the information retrieved.
Execution	3	I can come up with strategies to solve problems. I can provide safe and competent care when dealing with individual clients. I can apply communication skills to build a trusting nurse-patient relationship.
Lifelong Learning	4	I can connect academic learning to practical problems. I can apply the knowledge and skills I have learned to clinical practice and my personal life. I think I am a self-directed learner. I often think about how to improve my learning and problem-solving skills.

**Table 2 ijerph-19-15825-t002:** Characteristics of the three groups of nursing students (Chi-squared tests).

Group	Male, *n* (%)	Female, *n* (%)	X^2^	Sig.
Group 1	22 (37.9)	84 (31.8)	1.940	0.379
Group 2	16 (27.6)	95 (36.0)		
Group 3	20 (34.5)	85 (32.2)		

**Table 3 ijerph-19-15825-t003:** Comparison of the differences in students’ self-evaluations of core competencies between groups. (Paired-samples *t*-test).

Variable	*n*	Mean	SD	Mean of Difference	SD of Difference	95% C.I.		*t*	*p*-Value
						Lower	Upper		
Learning attitude									
Group 1 pre	106	3.7571	0.54167						
post	106	3.6958	0.62010						
				0.06132	0.69933	−0.07336	0.19600	0.903	0.369
Group 2 pre	111	3.8333	0.67026						
post	111	3.8446	0.53625						
				−0.01126	0.58035	−0.12043	0.09790	−0.204	0.838
Group 3 pre	105	4.0881	0.51653						
post	105	3.9286	0.57566						
				0.15952	0.59860	0.04368	0.27537	2.731	0.007
Problem identification									
Group 1 pre	106	3.2783	0.77791						
post	106	3.3538	0.72692						
				−0.07547	0.96558	−0.26143	0.11049	−0.805	0.423
Group 2 pre	111	3.4640	0.77960						
post	111	3.4955	0.69248						
				−0.03153	0.76062	−0.17460	0.11154	−0.437	0.663
Group 3 pre	106	3.8726	0.62129						
post	106	3.6934	0.70566						
				0.17925	0.76591	0.03174	0.32675	2.409	0.018
Information analysis									
Group 1 pre	106	3.6195	0.57171						
post	106	3.4874	0.70699						
				0.13208	0.73848	−0.01015	0.27430	1.841	0.068
Group 2 pre	111	3.7447	0.71627						
post	111	3.7297	0.60388						
				0.01502	0.62180	−0.10195	0.13198	0.254	0.800
Group 3 pre	106	4.0220	0.54872						
post	106	3.8113	0.59986						
				0.21069	0.61088	0.09304	0.32834	3.551	0.001
Execution									
Group 1 pre	106	3.7233	0.66424						
post	106	3.5535	0.69142						
				0.16981	0.74783	0.02579	0.31383	2.338	0.021
Group 2 pre	111	3.8228	0.66183						
post	111	3.8048	0.63161						
				0.01802	0.63220	−0.10090	0.13693	0.300	0.765
Group 3 pre	106	3.9937	0.59269						
post	106	3.8805	0.68265						
				0.11321	0.67281	−0.01637	0.24278	1.732	0.086
Life-long learning									
Group 1 pre	106	3.6840	0.65946						
post	106	3.5283	0.66576						
				0.15566	0.75786	0.00971	0.30161	2.115	0.037
Group 2 pre	111	3.6847	0.75431						
post	111	3.7027	0.64975						
				−0.01802	0.68234	−0.14637	0.11033	−0.278	0.781
Group 3 pre	106	4.0896	0.58268						
post	106	3.7901	0.75288						
				0.29953	0.67119	0.17026	0.42879	4.595	<0.000
Total scale									
Group 1 pre	106	3.6468	0.49657						
post	106	3.5454	0.57674						
				0.10142	0.61106	−0.01627	0.21910	1.709	0.090
Group 2 pre	111	3.7314	0.63063						
post	111	3.7365	0.52928						
				−0.00507	0.51020	−0.10104	0.09090	−0.105	0.917
Group 3 pre	105	4.0327	0.49404						
post	105	3.8304	0.58558						
				0.20238	0.51206	0.10328	0.30148	4.050	<0.000

**Table 4 ijerph-19-15825-t004:** Predictions of changes in the students’ self-evaluations of their core competencies (GEE model).

Dependent Variable	Parameter	*B*	S.E.	95% Wald C.I.		Wald X^2^	*p*-Value
				Lower	Upper		
Learning attitude	^1^ Group 3	0.223	0.0783	0.070	0.377	8.141	0.004
	^1^ Group 2	0.159	0.0749	0.013	0.306	4.532	0.033
	^2^ time 2	0.063	0.0665	−0.067	0.193	0.899	0.343
	^3^ Group 3 × time 2	0.084	0.0880	−0.089	0.256	0.904	0.342
	^4^ Group 2 × time 2	−0.084	0.0865	−0.254	0.085	0.949	0.330
Problem identification	^1^ Group 3	0.347	0.0954	0.160	0.534	13.226	0.000
	^1^ Group 2	0.134	0.0937	−0.050	0.317	2.035	0.154
	^2^ time 2	−0.057	0.0917	−0.237	0.122	0.392	0.531
	^3^ Group 3 × time 2	0.259	0.1188	0.026	0.492	4.763	
	^4^ Group 2 × time 2	0.040	0.1167	−0.189	0.269	0.116	
Information analysis	^1^ Group 3	0.352	0.0873	0.181	0.523	16.260	0.000
	^1^ Group 2	0.264	0.0873	0.093	0.435	9.171	0.002
	^2^ time 2	0.151	0.0705	0.013	0.290	4.605	0.032
	^3^ Group 3 × time 2	0.048	0.0911	−0.131	0.226	0.276	0.600
	^4^ Group 2 × time 2	−0.136	0.0917	−0.316	0.044	2.193	0.139
Execution	^1^ Group 3	0.294	0.0907	0.116	0.471	10.480	0.001
	^1^ Group 2	0.253	0.0855	0.085	0.421	8.739	0.003
	^2^ time 2	0.175	0.0707	0.037	0.314	6.142	0.013
	^3^ Group 3 × time 2	−0.024	0.0957	−0.211	0.164	0.061	0.805
	^4^ Group 2 × time 2	−0.160	0.0924	−0.341	0.021	3.002	0.083
Life-long learning	^1^ Group 3	0.262	0.0943	0.077	0.446	7.697	0.006
	^1^ Group 2	0.183	0.0863	0.014	0.352	4.506	0.034
	^2^ time 2	0.167	0.0724	0.025	0.309	5.339	0.021
	^3^ Group 3 × time 2	0.157	0.0971	−0.033	0.347	2.609	0.106
	^4^ Group 2 × time 2	−0.190	0.0969	−0.379	0.000	3.824	0.051
Total scale	^1^ Group 3	0.286	0.0766	0.136	0.436	13.902	0.000
	^1^ Group 2	0.199	0.0719	0.058	0.340	7.683	0.006
	^2^ time 2	0.120	0.0580	0.006	0.234	4.291	0.038
	^3^ Group 3 × time 2	0.103	0.0774	−0.049	0.254	1.758	0.185
	^4^ Group 2 × time 2	−0.131	0.0755	−0.278	0.017	2.993	0.084

Note: Group 1: Freshman students in 2013, Sophomore students in 2014. Group 2: Sophomore students in 2013; Junior students in 2014. Group 3: Junior students in 2013; Senior students in 2014.Time 1: 2013 data collection; time 2: 2014 data collection. Reference group: ^1^ = Group 1; ^2^ = time 1; ^3^ = Group 3 × time 1; ^4^ = Group 2 × time 1.

## Data Availability

Not applicable.

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
