# Peer review of "Effects of Problem-Based Learning Strategies on Undergraduate Nursing Students’ Self-Evaluation of Their Core Competencies: A Longitudinal Cohort Study"

_ijerph, 2022, doi:10.3390/ijerph192315825_

Round 1
Reviewer 1 Report
Dear Authors,
Thank you for the opportunity to review this interesting paper on the effects of the Problem-Based Learning Method on the competencies of undergraduate nursing students.
This work sheds light on the importance of PBL strategies. However, there are some main concerns that should be addressed before the official publication in the International Journal of Environmental Research and Public Health.
#Abstract
-
The authors should restructure the abstract by adding some background information to better understand the rationale behind this study and why it is crucial.
#Methods
-
This paragraph should be organized more consistently. Information considering participants is spread out in paragraphs 2.1., 2.3., and 3.1.. I would suggest the authors to create a participants section with all the characteristics to help readability and fluidity throughout the text.
-
Furthermore, a procedure paragraph might be added to appropriately describe when and how the self-reported questionnaire was administered and other salient information considering the study design.
-
2.2. Measures: The authors should provide more information about how the questionnaire was designed and the principles used in its implementation.
#Results
-
Before describing the results in paragraph 3.2., the authors should add a paragraph reporting an analytic plan of the statistical analysis. Which specific inferential tests have been used? Did the authors check the variables for homogeneity and linearity? Did they use parametric or non-parametric statistics?
-
Results should be reported more thoroughly, indicating the p-value and the specific tests used and other essential data that should be transparent.
#Discussion
-
The authors should introduce the discussion by adding initial phrases to sum up the objectives of this study and why it is important.
-
Some statements in the discussion part should be used with greater caution. The authors only used a self-reported questionnaire to measure the students' improvements over time. However, although self-reported measures are informative, they should be administered together with more objective measures (e.g., increase in scores during exams) to disentangle the effect of other possible intervening factors (e.g.-. the desirability of students to increase their abilities over time).
Previous research assessed single abilities and the evaluation of the students, however, these variables should also be considered when measuring self-reported abilities.
Author Response
Reviewer 1 |
Response |
Abstract: The authors should restructure the abstract by adding some background information to better understand the rationale behind this study and why it is crucial. |
Thank you for your comments. The following statements were added in the abstract section (p.1) Background: To response to the increasing patients' demands and to strengthen the capabilities of nursing professionals, nursing students are expected to develop problem-solving skills. Problem-based learning (PBL) is expected to provide effective simulation scenarios and realistic clinical conditions to help students achieve those learning goals. |
Method: This paragraph should be organized more consistently. Information considering participants is spread out in paragraphs 2.1., 2.3., and 3.1.. I would suggest the authors to create a participants section with all the characteristics to help readability and fluidity throughout the text. |
Thank you for your suggestions: “We grouped the students by their undergraduate academic level to study the effect of exposure to the PBL strategy: Group 1 (2013 as Freshman students, 2014 as Sophomore students) had PBL exposure of 1 year; Group 2 had PBL exposure of 2 years (2013 as Sophomore students, 2014 as Junior students); and Group 3 had PBL exposure of 3 years (2013 as Junior students, 2014 as Senior students).” were moved from 2.3 to 2.1. Sample size information from 3.1 was included in 2.1 to provide more comprehensive participant information. |
Method: Furthermore, a procedure paragraph might be added to appropriately describe when and how the self-reported questionnaire was administered and other salient information considering the study design. |
Thank you for your suggestions: We briefly describe the recruitment process in section 2.1 (p.3) but to make it clearer to the readers, we inserted the following section to section 2.2: “Students (from freshman to junior classes) were recruited to participate in the survey in September 2013. Those students were tracked to complete the survey in September 2014. Overall, a total of 322 nursing students completed the same questionnaire in both 2013 and 2014.” |
Method: 2.2. Measures: The authors should provide more information about how the questionnaire was designed and the principles used in its implementation. |
Thank you for your suggestions: The following statements were inserted in section 2.2: “Content validity was performed and five dimensions were identified: learning attitude, problem identification, information analysis, execution, and life-long learning. The overall content validity index of the instrument was high (overall S-CVI=0.93) The final instrument include sixteen items out of twenty-four items originally developed.” |
Results: Before describing the results in paragraph 3.2., the authors should add a paragraph reporting an analytic plan of the statistical analysis. Which specific inferential tests have been used? Did the authors check the variables for homogeneity and linearity? Did they use parametric or non-parametric statistics? |
Thank you for your suggestions: Paired-samples t-test, and Generalized estimating equations were used to analyze data. We consulted a statistician to make sure all the analytical methods were satisfactory. |
Results: Results should be reported more thoroughly, indicating the p-value and the specific tests used and other essential data that should be transparent. |
Thank you for your comments: P values were all provided. Specific tests used (mentioned in the original manuscript) but tests were added in table captions in the revision. Table 1: Chi-square tests Table 2: Paired-sampled t-tests were used. Table 3: GEE model
|
Discussion: The authors should introduce the discussion by adding initial phrases to sum up the objectives of this study and why it is important. |
Thank you for your suggestions: The following statements were added: (p. 5) The study aimed to explore the effects of PBL strategies on nursing students’ self-evaluation of core competencies. It was imperative for nursing educators to evaluate the effects of PBL teaching strategies on students. These results of the study would help nursing educators to develop curricula and improve teaching strategies. |
Discussion: Some statements in the discussion part should be used with greater caution. The authors only used a self-reported questionnaire to measure the students' improvements over time. However, although self-reported measures are informative, they should be administered together with more objective measures (e.g., increase in scores during exams) to disentangle the effect of other possible intervening factors (e.g.-. the desirability of students to increase their abilities over time). |
Thank you for your comments.
We agree with the reviewer’s opinions. We also recognized the limitations of the self-reported measures and will conduct a follow-up study with multiple measures to evaluate the effect of PBL on students.   |
Discussion: Previous research assessed single abilities and the evaluation of the students, however, these variables should also be considered when measuring self-reported abilities. |
Thank you for your comments. We agree with the reviewer’s opinions.
We recognized all the scientific and practical contributions of the previous studies. In our discussion section, we tried to highlight the contributions of our study to nursing education despite the limitations of using only the self-reported measure. |

Reviewer 2 Report
This article is focused on “to explore the effects of problem-based learning (PBL) strategies on nursing students’ self-evaluation of core competencies”.
Comments and suggestions
Some minor format issues…
When referring to some articles it appears ‘og’… It should be ‘and’ or ‘&’?
--
Measures
The different items analyzed in this research [learning attitude (four items), problem identification (two items), information analysis (three items), execution (three items), and life-long learning (four items)] are not described in the article. All the different items should be defined in the article, at least in an appendix.
Data collection
The introduction gives review about PBL, referencing quite recent research articles. However, in your research, you are presenting data from 2013 and 2014.
Have you continued surveying students about the research question of this study? If this is the case, adding these new data would enrich this study.
-
Tables and Figure is not included in the article …
There are no tables and figures included in the article, despite are mentioned in the text of the article (Table 1; Table2; Table 3; Figure 1). Tables and Figure should be included to properly review the results of the article.
-
Limitations
In the article you state: “Moreover, we only evaluated the effectiveness of the PBL learning strategy over two years. Outcome assessments could be conducted in subsequent years. Extending the time for data collection would enable an analysis of the trajectory of the long-term effects and provide more specific data on the effectiveness of PBL.”
In the research work you are presenting data from 2013 and 2014. Do you have more recent data (2015, 2016... 2022) to cope to this limitation that you identify?
References
Some format issues...
· He, Y., Du, X., Toft, E., Zhang, X., Qu, B., Shi, J., . . . Zhang, H. (2018). A Comparison between the Effectiveness of PBL and LBL on Improving Problem-Solving Abilities of Medical Students Using Questioning. Innovations in Education and Teaching Interna-tional, 55(1), 44-54. Sótt af http://search.ebscohost.com/login.aspx?direct=true&db=eric&AN=EJ1171223&lang=zh-tw&site=ehost-live
http://dx.doi.org/10.1080/14703297.2017.1290539
· Kwan, C.-Y. (2017) Problem-Based Learning in Medical Education in Taiwan: Observations and a Commentary ???????????????????????. Journal of Medicine and Health 6(1), 1-11.
Kwan, C.-Y. og Lee, M.-C. (2018). From Problem-Based Learning in Classrooms to Holistic Health Care in Workplaces with Special Emphasis in Chinese Societies ????????????????????. Journal of Medicine and
Rico, R. og Ertmer, P. A. (2015). Examining the Role of the Instructor in Problem-Centered Instruction %J TechTrends: Linking Re-search and Practice to Improve Learning. 59(4), 96-103.
Doi is missing ... DOI:10.1007/s11528-015-0876-4
Shin, I.-S. og Kim, J.-H. (2013). The Effect of Problem-Based Learning in Nursing Education: A Meta-Analysis. Advances in Health Sciences Education, 18(5), 1103-1120. Sótt af http://search.ebscohost.com/login.aspx?di-rect=true&db=eric&AN=EJ1036017&lang=zh-tw&site=ehost-live
http://dx.doi.org/10.1007/s10459-012-9436-2
Author Response
Reviewer 2 |
Response |
Some minor format issues…
When referring to some articles it appears ‘og’… It should be ‘and’ or ‘&’? |
Thank you for your comments. Those formatting errors were caused by Endnote conversion and were corrected. |
Measure: The different items analyzed in this research [learning attitude (four items), problem identification (two items), information analysis (three items), execution (three items), and life-long learning (four items)] are not described in the article. All the different items should be defined in the article, at least in an appendix. |
Thanks for your comments: The following descriptions were added to the manuscript: “Learning attitude” was defined as students’ tendency to respond a certain way towards PBL learning. Four items were included in the questionnaire to measure students’ learning attitude. “Problem identification” was students’ ability to identify the key problem to be solved or addressed. Two items were used to assess subjects’ problem-identification skills. “Information analysis” aimed to check students’ ability to inspect and comprehend information collected. Three items were included to evaluate students’ information analysis competency. “Execution” focused on students’ abilities to perform a certain task or complete study goals. Three items were employed to assess students’ execution competency. “Life-long learning” was described as students’ self-initiated and ongoing learning intentions or behaviors. It was measured by 4-item questions.” (p.3) |
Data Collection: The introduction gives review about PBL, referencing quite recent research articles. However, in your research, you are presenting data from 2013 and 2014.
Have you continued surveying students about the research question of this study? If this is the case, adding these new data would enrich this study. |
Thanks for your comments: Our original IRB only approved us to collect data from 2013 & 2014. Additional IRB approval has to be applied before we can continuously collect more data. Next, we plan to conduct a follow-up study to track the long-term effects of PBL on those competencies of our graduates. |
Tables and Figures: There are no tables and figures included in the article, despite are mentioned in the text of the article (Table 1; Table2; Table 3; Figure 1). Tables and Figure should be included to properly review the results of the article. |
Thanks for your comments: We apologize for the miss tables and figures. Originally, we submitted tables and figure separately. In the revision, we have included them in the manuscript. |
Limitations: In the article you state: “Moreover, we only evaluated the effectiveness of the PBL learning strategy over two years. Outcome assessments could be conducted in subsequent years. Extending the time for data collection would enable an analysis of the trajectory of the long-term effects and provide more specific data on the effectiveness of PBL.”
In the research work you are presenting data from 2013 and 2014. Do you have more recent data (2015, 2016... 2022) to cope to this limitation that you identify? |
Thanks for your suggestions--- We plan to conduct a follow-up research and the following statement “A follow-up study will be conducted to examine the long-term effects of PBL on the core competencies of graduates.” was added to the manuscript. (p.7) After the follow-up study is completed, we will summarize and report our findings.
|
References: Some format issues... |
Those formatting errors were corrected. Thank you for your comments. |

Round 2
Reviewer 1 Report
Dear Authors,
A great part of my comments was adequately addressed. I think that the current version of the manuscript benefitted from the various comments provided by the different reviewers and I would recommend this work for publication.
Author Response
Thank you so much for your kind comments!
And again, we greatly appreciate your previous suggestions to make this revision a stronger version.

Reviewer 2 Report
The revised version has improved the structure of the article.
Comments and suggestions
· Minor format issues
o Abstract.- Problem-based learning (PBL) is expected t0 provides effective simulation -> to
o Abstract.- You can delete explicit references to p values (numbers) within the abstract.
o Abstract.- You can delete the sections in the format that are written in the abstract. E.g. Instead of ‘Objective:’ -> The objective of this article is ; Instead of ‘Results’ -> The results that; …
o Abstract.- groups -- group 1 (one-year PBL exposure), group 2 (two-year PBL exposure), and group 3 (three-year exposure).
o Epigraph 4.1.- It should not be written in capital letters (4.1. RESEARCH LIMITATIONS)
Additional considerations
· The different items analyzed in this research [learning attitude (four items), problem identification (two items), information analysis (three items), execution (three items), and life-long learning (four items)] should be listed, at least, in an appendix.
Eg. The four items related to ‘learning attitude’ are: (1) xxx; (2) yyy; (3) kkk; and (4 iiii). It could be also shown in atable.
· Table 2 & Table 3 has been added at the end of the article. You have to options: (1) move each one of this tables below the first paragraph that each table is mentioned; (2) include and appendix at the end of the text, following the format rules (moving it to the right position in the article, and also in terms of names of the tables; you can look at it in ‘template’ included in ‘Instructions for authors’ -in the site of the review-).
· Figure 1.- you should follow the format rules; you can look at it in ‘template’ included in ‘Instructions for authors’ -in the site of the review-).
· Conclusions. - It seems too synthetic and a little bit decontextualized. Consider moving to this section the first two and a half lines included in epigraph 4.1. This contend is not required in subsection 4.1, while contextualizes your research when you include these lines in the last section (Conclusions). Including a very brief reference to your future plans would add consistency to the section.
Author Response
Minor format issues o Abstract.- Problem-based learning (PBL) is expected t0 provides effective simulation -> to o Abstract.- You can delete explicit references to p values (numbers) within the abstract. o Abstract.- You can delete the sections in the format that are written in the abstract. E.g. Instead of ‘Objective:’ -> The objective of this article is ; Instead of ‘Results’ -> The results that; … l Abstract.- groups -- group 1 (one-year PBL exposure), group 2 (two-year PBL exposure), and group 3 (three-year exposure). l Epigraph 4.1.- It should not be written in capital letters (4.1. RESEARCH LIMITATIONS) |
Thank you for your comments-- those formatting problems were corrected. |
The different items analyzed in this research [learning attitude (four items), problem identification (two items), information analysis (three items), execution (three items), and life-long learning (four items)] should be listed, at least, in an appendix. Eg. The four items related to ‘learning attitude’ are: (1) xxx; (2) yyy; (3) kkk; and (4 iiii). It could be also shown in atable.
|
Thank you for your suggestions-- we add a new table to list all 16 items in the survey (Table 1). |
Table 2 & Table 3 has been added at the end of the article. You have to options: (1) move each one of this tables below the first paragraph that each table is mentioned; (2) include and appendix at the end of the text, following the format rules (moving it to the right position in the article, and also in terms of names of the tables; you can look at it in ‘template’ included in ‘Instructions for authors’ -in the site of the review-). · Figure 1.- you should follow the format rules; you can look at it in ‘template’ included in ‘Instructions for authors’ -in the site of the review-). |
Thank you for your suggestions—We have moved the tables to the main texts. We had used the tracking change function for you to track those modifications. |
Conclusions. - It seems too synthetic and a little bit decontextualized. Consider moving to this section the first two and a half lines included in epigraph 4.1. This contend is not required in subsection 4.1, while contextualizes your research when you include these lines in the last section (Conclusions). Including a very brief reference to your future plans would add consistency to the section. |
Thank you for your suggestions—We have removed 4.1 section and integrated contents from section 4.1 into the conclusion (as the following). |
